# Femtosecond Laser Microfabrication of Artificial Compound Eyes

**Fan Zhang [1], Huacheng Xu [2], Qing Yang [3], Yu Lu [2], Guangqing Du [2] and Feng Chen [2,\***

[1] School of Physics Science and Information Technology, Liaocheng University, Liaocheng 252000, China; zhangfan@lcu.edu.cn
[2] State Key Laboratory for Manufacturing System Engineering and Shaanxi Key Laboratory of Photonics Technology for Information, School of Electronic Science and Engineering, Xi'an Jiaotong University, Xi'an 710049, China; huacheng.xu@stu.xjtu.edu.cn (H.X.); luyu90@xjtu.edu.cn (Y.L.); guangqingdu@mail.xjtu.edu.cn (G.D.)
[3] School of Instrument Science and Technology, Xi'an Jiaotong University, Xi'an 710049, China; yangqing@mail.xjtu.edu.cn
\* Correspondence: chenfeng@xjtu.edu.cn

**Abstract:** Over millions of years of evolution, arthropods have intricately developed and fine-tuned their highly sophisticated compound eye visual systems, serving as a valuable source of inspiration for human emulation and tracking. Femtosecond laser processing technology has attracted attention for its excellent precision, programmable design capabilities, and advanced three-dimensional processing characteristics, especially in the production of artificial bionic compound eye structures, showing unparalleled advantages. This comprehensive review initiates with a succinct introduction to the operational principles of biological compound eyes, providing essential context for the design of biomimetic counterparts. It subsequently offers a concise overview of crucial manufacturing methods for biomimetic compound eye structures. In addition, the application of femtosecond laser technology in the production of biomimetic compound eyes is also briefly introduced. The review concludes by highlighting the current challenges and presenting a forward-looking perspective on the future of this evolving field.

**Keywords:** femtosecond laser; compound eyes; microlens arrays; wet etching; dry etching

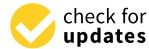



## 1. Introduction

Compound eyes represent an exquisitely sophisticated imaging system in nature, prevalent among insects and crustaceans in the natural world. They are composed of thousands of tiny ommatidia, mainly hexagonal and circular in shape, each of which acts as an independent photoreceptor unit (Figure 1a) [1]. The basic structure of an individual ommatidium includes the corneal lens, crystalline cone, and rhabdom bundle [2–5], as illustrated in Figure 1b [6]. Different species exhibit distinct compound eye structures, broadly categorized into two types: apposition compound eyes and superposition compound eyes. Most diurnal insects, shallow-water organisms, and crustaceans possess apposition compound eyes, where each ommatidium receives light from a single lens. In contrast, superposition compound eyes are more suited for nocturnal animals [6].

In apposition compound eyes, each optical channel is strategically isolated from its adjacent channels, mitigating contrast degradation caused by stray light and minimizing interference [7,8]. Within each ommatidium, a lens facilitates the passage of light signals to the photoreceptor region [9], giving rise to visual perception (Figure 2a) [6]. The light-receiving portion comprises retinal cells known as the rhabdom. The primary focusing element, a crystalline cone, is positioned between the cornea and the rhabdom. Images captured by these compound eyes undergo parallel processing, with each corneal lens adept at simultaneously transmitting and receiving signals. This capability facilitates swift motion

detection and image recognition. Nevertheless, a drawback is the notable reduction in brightness. Despite the relatively limited individual field of view for each ommatidium, the coordinated efforts of multiple ommatidia contribute to extensive visual perception, even achieving a complete 360° field of view. Each ommatidium possesses imaging capabilities, enabling arthropods to gather ample information in complex environments. This distinctive structure is characterized by a large field of view, minimal distortion, and rapid detection of moving objects [10–14]. Apposition compound eyes typically comprise thousands of irregularly arranged hexagonal ommatidia, a prevalent feature in structures observed in dragonflies and bees.

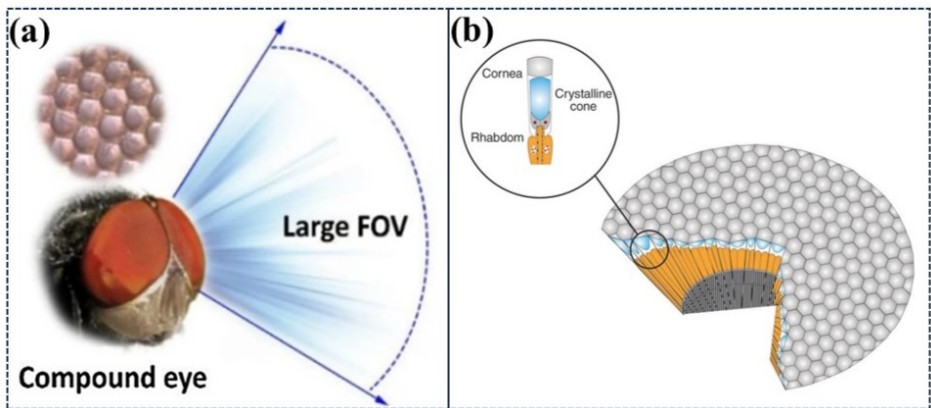

**Figure 1.** (**a**) Working principles and properties of a compound eye. Figure reproduced from ref. [1], American Chemical Society. (**b**) Schematics of compound eye. Figure reproduced from ref. [6], Science.

Superposition compound eyes are primarily found in nocturnal animals and deep-sea crustaceans, specifically engineered to efficiently absorb light from diverse directions. The rhabdom within each ommatidium can capture two or more light signals from the corneal lens, resulting in superior focusing ability and a larger numerical aperture compared to corresponding compound eyes, as shown in Figure 2b [6]. In addition to these advantages, superposition compound eyes also demonstrate exceptional light sensitivity, although they may encounter blurriness at the edges of the imaging field. Typically, each channel of a superposition compound eye transmits only a segment of the extensive visual field image. Within superposition compound eyes, the combination of multiple summed signals and individual signals collectively constructs an overall image [15,16]. The intricate design and challenging manufacturing processes pose significant obstacles to the construction of artificial superposition compound eye imaging systems. In the realm of biomimicry, apposition-type compound eyes are more commonly adopted.

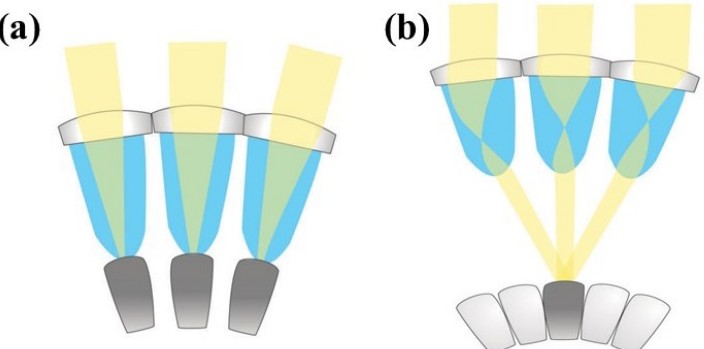

**Figure 2.** (**a**) Light flow through an apposition compound eye. (**b**) Light flow through a superposition compound eye. Figure reproduced from ref. [6], Science.

With the evolving requirements of modern optical systems and devices towards lightweight, miniaturized, arrayed, integrated, three-dimensional, and functionalized designs, traditional single-eye systems are constrained by factors such as size, functionality, field of view, and resolution, making them inadequate to meet the demands of optical system development and practical applications. Inspired by the structure of arthropods' compound eyes in the natural world, researchers have utilized micro-nanofabrication techniques to create optical microstructures based on planar and curved bio-inspired compound eye structures [17–21]. Due to the advantages of high integration, small size, large field of view imaging, and minimal impact from diffraction limits, biomimetic compound eyes find widespread applications in the military (missile guidance and radar detection), industrial (micro cameras and robot vision), and clinical medical (endoscopy) fields [22–34].

Presently, researchers have predominantly categorized biomimetic compound eye structures into two types based on their geometric characteristics: planar compound eye structures and curved compound eye structures. Planar compound eye structures are characterized by thousands of closely interconnected planar micro-lenses, densely arranged and uniformly distributed on a planar substrate [35–38]. In contrast, curved compound eye structures emulate the architecture of biological compound eyes, featuring a closely connected array of hexagonal micro-lenses arranged on a curved substrate [2,7,39,40]. As micro-optical systems continue to advance and the utilization of compound eye structures becomes more prevalent, there is a growing emphasis on exploring micro-nanofabrication techniques. This exploration aims to achieve higher integration and enhanced functionality of planar compound eye structures. Simultaneously, there is a concerted effort to develop efficient methods for preparing high-quality curved compound eye structures. This dual focus on advancing both planar and curved structures reflects the current forefront of development and research in the field of biomimetic compound eyes.

In recent years, as micro-nanomanufacturing technology has advanced continuously, a multitude of innovative microfabrication processes have been gradually integrated into the preparation of a microlens array (MLA) [41–46]. The current repertoire of fabrication methods for planar compound eye structures encompasses diverse techniques such as self-assembly, photolithography, thermal reflow, surface energy-driven methods, and micro-spray printing. Self-assembly relies on the inherent interactions among atoms, ions, and molecules to spontaneously aggregate and bond, resulting in the formation of numerous structurally stable micro-/nanostructures with regular shapes. These structures are then strategically arranged in a specific pattern to yield an MLA [47–50]. Photolithography and thermal reflow techniques involve the initial formation of a micro-cylinder array using photolithography. Subsequently, by heating the photolithography material to the glass transition temperature, the micro-cylinders are transformed into spherical microlens structures under the influence of surface tension [51,52]. As the exploration of artificial biomimetic compound eye structures has progressed, there has been a successive unveiling of curved biomimetic artificial compound eye MLAs. These advancements incorporate various materials and technological approaches, including lithography [53], three-dimensional reconstruction [2], surface wrinkling technology [54], etc. While traditional fabrication methods exhibit certain feasibility, they are confined by processing accuracy, particularly in achieving flexible three-dimensional curved surface structures. This necessitates the adoption of more intricate processing procedures.

The femtosecond laser, characterized by an ultra-short pulse duration ranging from $10^{-14}$ to $10^{-15}$ s, represents a revolutionary advancement in laser technology. While a single femtosecond laser pulse carries energy in the range of tens of microjoules to a few millijoules, its distinctiveness lies in the remarkably short pulse duration, leading to an exceptionally high peak power. The ultra-short pulse width of femtosecond lasers plays a pivotal role in effectively circumventing physical and chemical processes that occur in picoseconds or longer durations, with thermal diffusion being the most notable. Traditional laser processing approaches often result in a substantial thermal diffusion zone, adversely impacting processing accuracy. In contrast, femtosecond laser micromachining introduces

a "cold" processing approach that minimizes damage to the surrounding area and facilitates ultra-high precision micro-/nanofabrication [55–57]. Therefore, femtosecond laser micromachining technology not only fulfills the demands for flexible three-dimensional processing of devices, but also achieves downsizing [58–65], establishing itself as a potent tool for the fabrication of microlenses in compound eyes.

This article provides a comprehensive overview of the advancements in the femtosecond laser processing of microlenses for compound eyes, classifying the progress into additive manufacturing and subtractive manufacturing. Initially, a succinct introduction to the working principles of biological compound eyes lays the foundation for designing biomimetic compound eyes. The subsequent sections delve into a detailed exploration of crucial manufacturing methods employed for crafting biomimetic compound eye structures. To conclude, the challenges and prospects of MLA manufacturing in both industrial and academic research are addressed.

## 2. Femtosecond Laser Processing Technology

### 2.1. Two-Photon Polymerization Technology

Two-photon polymerization is a captivating phenomenon arising from the interaction of light and matter. Ordinarily, a molecule or atom absorbs a single photon, undergoing a transition from the ground state to the excited state. However, when exposed to high-energy lasers, materials such as photoresists or organic substances exhibit multi-photon transitions, reaching higher energy levels and initiating cross-linking in photosensitive polymer monomers [66]. This technology harnesses femtosecond lasers for ultra-fine processing. It capitalizes on the two-photon absorption phenomenon induced by femtosecond lasers at the focus to solidify the photoresist material. In areas with insufficient laser intensity for two-photon absorption, the material remains in a liquid state. Precise control over the preparation of three-dimensional structures is achieved by manipulating the movement of the laser focus, facilitating genuine three-dimensional, high-precision processing [67–70]. Utilizing femtosecond laser pulses and two-photon absorption, this manufacturing method reduces the feature size of 3D printing to sub-micrometer levels, aligning with the prevailing trend towards miniaturization.

Femtosecond laser two-photon polymerization technology stands out as a versatile approach for crafting three-dimensional structures with nanoscale precision in diverse geometric forms. Noteworthy examples of its application encompass the fabrication of complex 3D structures, such as micro-bull, nano-oscillators, and photonic crystals [67,71,72]. Guo et al., applied femtosecond two-photon polymerization for the manufacturing of microlenses, employing a continuous layer-by-layer thickness annular scanning pattern [73]. The optimization of crucial process parameters influencing the quality of the resulting microlenses, including scanning mode and step size, was achieved through a combination of theoretical simulations and experimental trials. Figure 3a illustrates the successful production of a 2 × 2 MLA with a diameter of 15 μm. Due to limitations in the processing technology, most MLAs have a low filling factor, resulting in relatively low optical efficiency. To address this issue, Wu et al., pioneered the fabrication of high-quality hexagonal MLAs by capitalizing on the self-smoothing effect and employing an equidistant arc scanning technique, as illustrated in Figure 3b,c [74]. The hexagonal configuration was selected due to its close approximation to a circular shape, minimizing off-axis deviations in incident light and consequently reducing spherical aberration during imaging. This MLA achieved a filling factor of nearly 100% [Figure 3d], and its exceptional optical performance garnered acclaim for delivering clear focus and high-resolution imaging.

Benefiting from its meticulous point-by-point processing capability, two-photon polymerization has emerged as a versatile technique for crafting intricate three-dimensional microstructures [75]. This groundbreaking approach addresses the inherent constraints associated with the repetitive and uniformly curved nature of conventional compound eye lens arrays, holding significant promise for enhanced optical performance. Sun et al., introduced a novel concept of microlens array with different curvatures (MLADC) [76],

employing two-photon polymerization technology to fabricate microlenses with variable focus, as shown in Figure 4a. The resulting microlenses exhibited diverse heights and curvatures, facilitating the imaging of objects at different positions. The ability to adjust curvature enables comprehensive scene imaging. In the realm of commercial image detectors, nearly all devices are flat due to manufacturing limitations, necessitating multiple lenses in complex optical systems to achieve a flat image plane [77]. MLADC adeptly tackles this challenge, featuring unit lenses in the central region that are taller and possess a shorter focal length than those at the periphery, as illustrated. Unlike the flat focal planes of traditional MLAs [Figure 4b,d], MLADC introduces a curved focal plane [Figure 4c,e], providing a mechanism to compensate for the inherent field curvature. In comparison to traditional MLAs, MLADC exhibits a distinctive optical performance, playing a pivotal role in optimizing optical system structures, streamlining optical components, and particularly addressing field curvature corrections.

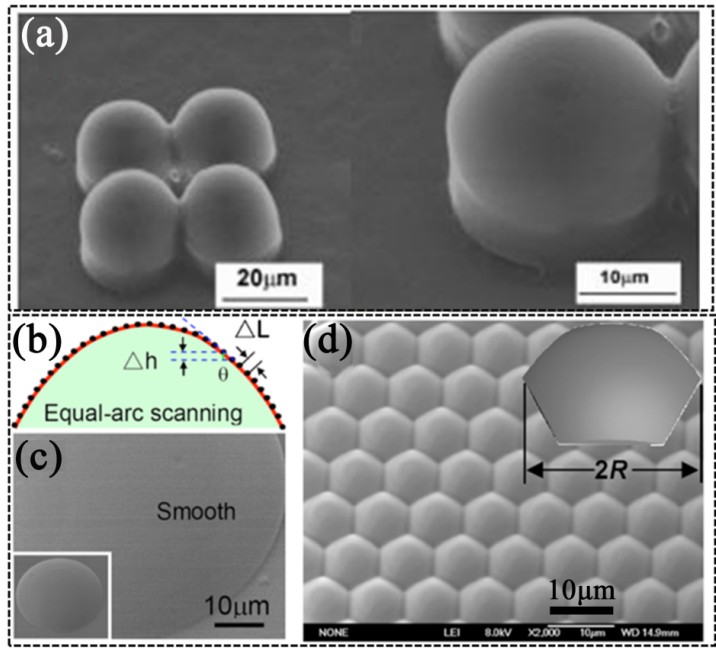

**Figure 3.** (**a**) The SEM of 2 × 2 array microlens and its close-up view. Figure reproduced from (**a**) ref. [73], Optical Society of America. (**b**) Equal-arc scanning mode. (**c**) SEM image of a single microlens. (**d**) Partially enlarged image in a 35° flat view. Figure reproduced from (**b**–**d**) ref. [74], American Institute of Physics.

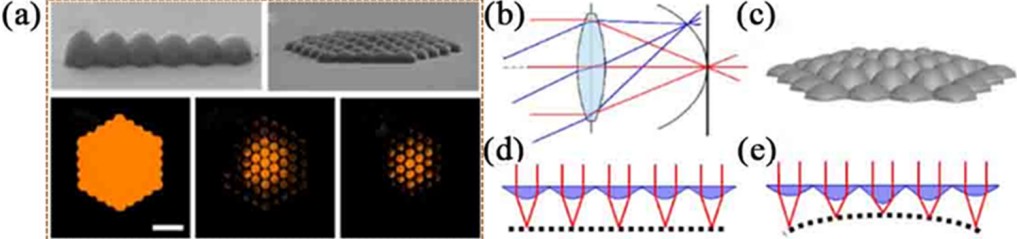

**Figure 4.** (**a**) The SEM of MLADC. Note the height differences of lenses. (**b**) Schematic of field curvature. The red and blue lines represent different fields of view lights, illustrating their inability to be focused by a lens on a flat plane, as opposed to a curved plane. (**c**) Prespective view of the MLADC. (**d**,**e**) Ordinary MLA and MLADC. The difference between conventional MLAs and MLADC lies in the focal plane; the former(**d**) has a flat focal plane, while the latter (**e**) has a curved focal plane. Figure reproduced from (**a**–**d**) ref. [76], Optical Society of America.

The innovation in two-photon polymerization technology lies in its ability to construct MLAs on curved substrates without the need for masks or templates, mitigating the risk of deformation during the transition from a planar to a curved structure. In a notable study by Wu et al., femtosecond laser pixel modulation with high-speed scanning was employed to process SU-8 material, resulting in the creation of biomimetic compound eye structures [78], as is visually represented in Figure 5a–d. In the initial phase, a laser beam with a pixel spacing of 400 nm × 400 nm and a power of 15 mW was used to scan the photoresist, forming a curved, large base. The entire processing procedure is illustrated in Figure 5e. Subsequently, a laser beam with a power of 6 mW and a pixel spacing of 100 nm × 100 nm was employed to process the small eye array on the curved substrate. This processing technique ensures meticulous control over the shape, height, size, profile, and curvature of the spherical large base through optimized program design and threshold laser power. By strategically controlling the height of the spherical large base at 5.3 μm, 10.7 μm, and 16.5 μm, the study achieved undistorted wide-angle imaging at 30°, 60°, and 90° angles.

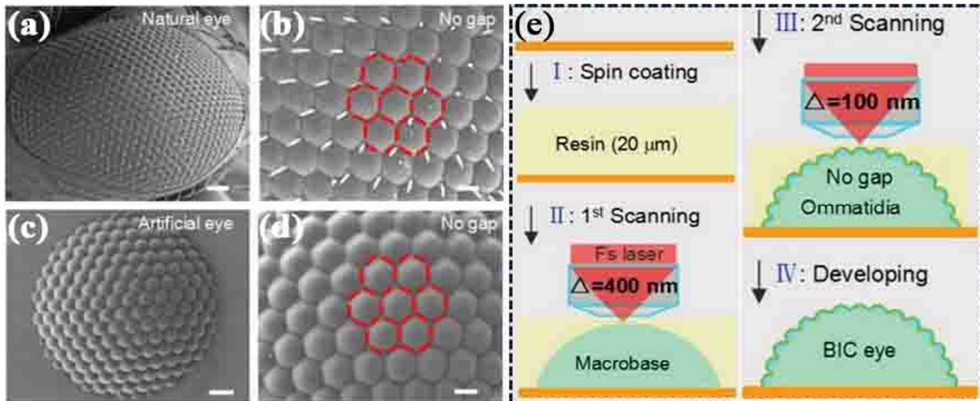

**Figure 5.** (**a**) SEM top-view images of the natural compound eye and (**c**) the artificially prepared compound eye both exhibit a 100% filling of small eye structures (**b**,**d**). The scale bars for (**a**–**d**) are 40, 10, 10, and 5 μm, respectively. (**e**) Schematic diagram of the artificial compound eye processing modes: I photoresist samples were prepared by spin coating SU-8 films on microscope cover slides, II illustrates the processing of a hemispherical base, III demonstrates the processing of densely distributed small eyes on a spherical surface, and IV represents the final composite structure. Figure reproduced from (**a**–**e**) ref. [78], Wiley-VCH.

Femtosecond laser two-photon polymerization is widely employed with photosensitive polymers such as photoresists, proteins, liquid crystals, and soft polymers [79,80]. Leveraging the stimuli-responsive characteristics of polymer materials, researchers have successfully fabricated tunable biomimetic compound eye lenses. A notable example is the work by Ma et al., who developed a tunable biomimetic compound eye lens based on bovine serum albumin (BSA). This compound eye features a double-layer structure with an SU8 photoresist core and a shell made of BSA protein [81], as shown in Figure 6b–d. Bovine serum albumin, being highly responsive to the surrounding pH environment, exhibits contraction and expansion at different pH values [82,83]. Exploiting the pH-responsive properties of BSA protein, dynamic adjustments of the field of view and focal length were achieved by altering the pH of the surrounding solution. The field of view angle varied from 35° (pH = 5) to 80° (pH = 13), and changes in the microlens shape resulted in variations in focal length. Consequently, under different pH conditions, the lens could effectively detect targets at different distances, as shown in Figure 6a. This tunable compound eye introduces a novel perspective for adjustable imaging and holds promising applications in the fields of micro-robotic vision, optofluidic devices for wide-field monitoring, medical endoscopic diagnosis, and microscale particle image velocimetry.

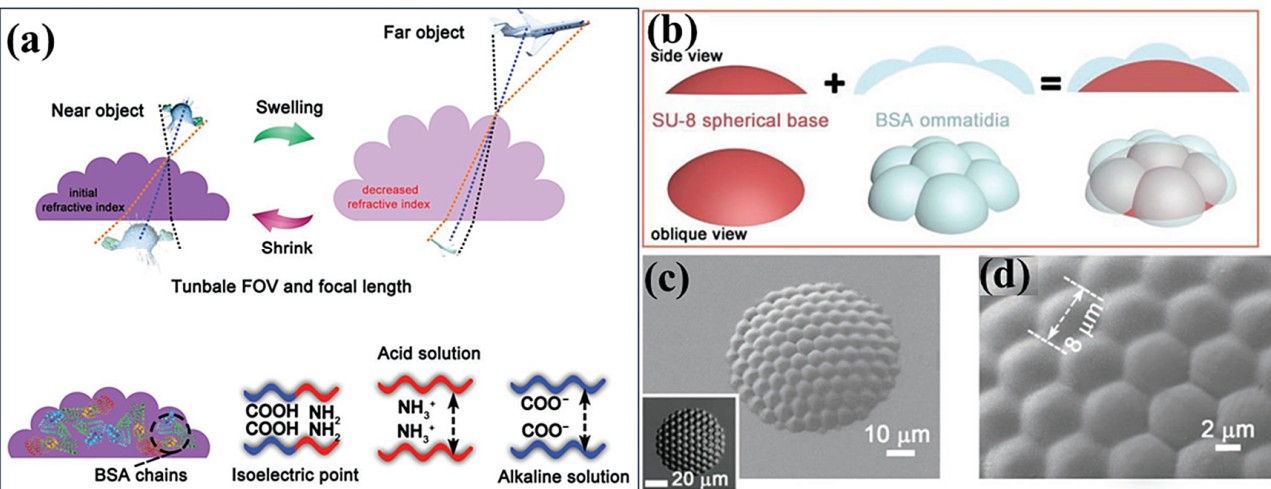

**Figure 6.** (**a**) The schematic of the reversible swelling and contraction characteristics of the protein compound eye, enabling adjustment of the field of view and focal length. The diagram below depicts the potential mechanism of the adjustable protein compound eye. (**b**) The fabrication of the SU-8/BSA-based compound eye. First, an SU-8 microlens was fabricated; subsequently, the ommatidia based on BSA were integrated onto the SU-8 microlens through a secondary femtosecond laser direct writing process. (**c**) SEM images of the compound eye based on BSA. Figure reproduced from (**a**–**d**) ref. [81], Wiley-VCH.

Two-photon polymerization technology is a suitable tool for manufacturing complex multi-lens optical systems with high optical performance and significant compactness. Gissibi et al., utilized femtosecond laser two-photon lithography to fabricate a multilayer MLA [75]. This integrated three-dimensional microlens exhibited excellent surface quality and imaging performance, enhancing the integration of micro-optical systems. Non-spherical lenses were employed to compensate for optical aberrations, improving the imaging quality of each surface. Subsequently, composite lenses were manufactured using a femtosecond laser lithography system. Several different single lenses were combined in a supporting shell to form a composite lens, meeting various requirements simultaneously. The simulation of the USAF 1951 resolution test chart in Figure 7a confirms these observations. Single-layer lenses exhibit strong field-related aberrations, which can be identified from the blurred edges of the image. Additionally, the magnification of single-layer lenses is uneven across the entire image field. In Figure 7a, this barrel distortion is clearly visible at the edges of the image. Figure 7b shows SEM images of the single-layer, double-layer, and triple-layer lenses. Figure 7c illustrates the optical performance of single-layer, double-layer, and triple-layer lenses in imaging the USAF 1951 resolution test chart. This work lies at the intersection of micro-optics and nano-optics, representing a paradigm shift in micro-optics. The entire process takes only a few hours, from lens design to production and testing to the final functioning optical device.

The parameters for processing compound eyes in several works are compared in Table 1. It can be observed that femtosecond laser two-photon polymerization technology can achieve high-precision manufacturing at the micro and even nano scales, providing unique advantages in the field of micro-/nanofabrication. However, this technology has a relatively low processing efficiency, especially unsuitable for the preparation of large-area structures. Additionally, its applicability is limited to specific materials, mainly photoresists or organic materials. The soft material and poor mechanical strength and thermal stability of photoresist materials significantly restrict the practical application of this technology in production.

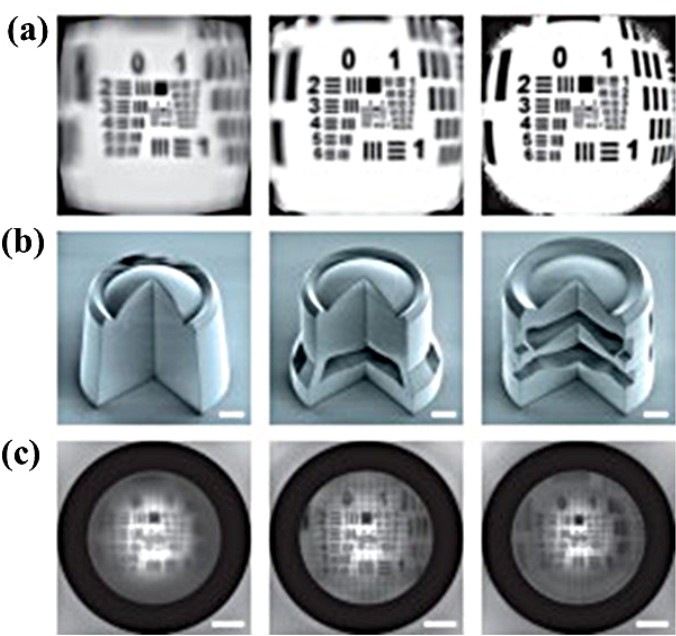

**Figure 7.** (**a**) Simulated image of the USAF 1951 resolution test chart. (**b**) The SEM images of single-layer, double-layer, and triple-layer lenses. (**c**) The image of the lens imaging the USAF 1951 resolution test chart. The distance between the lens and the target is 20 mm, scale: 20 μm. Figure reproduced from (**a**–**c**) ref. [75], Nature.

**Table 1.** Parameters of compound eyes fabricated using two-photon polymerization technology.

| Year | No. of Ommatidium | Ommatidium Diameter (μm) | Ommatidium Height (μm) | FOV (°) | Laser Parameters | Materials | References |
|---|---|---|---|---|---|---|---|
| 2006 | 4 | 15 | 8 | - | 800 nm, 80 fs, 80 MHz | commercial resin | [74] |
| 2010 | ~91 | 10 | 1.2–2.4 | - | 800 nm, 120 fs, 82 MHz | SU-8 | [75] |
| 2015 | ~67 | 20 | 3–10 | - | 800 nm, 120 fs, 82 MHz | SU-8 | [77] |
| 2014 | ~150 | 16 | 4 | 30–90 | 790 nm, 120 fs, 80 MHz | SU-8 | [79] |
| 2019 | ~93 | 8 | 2 | 35–80 | 800 nm, 120 fs, 80 MHz | SU-8 | [82] |
| 2016 | - | 100–200 | 115 | 80 | - | photoresist | [76] |

### 2.2. Femtosecond Laser Ablation Processing Technology

With the continuous advancement of femtosecond laser technology, femtosecond laser ablation processing has emerged as an innovative and effective technique for crafting intricate three-dimensional micro-/nanostructures. This method capitalizes on the ultra-high peak power of femtosecond lasers and the attributes of "cold" processing, making it suitable for high-hardness materials like diamonds and sapphires [84–87]. Its widespread application in creating complex structures demonstrates its efficacy. The interaction between the femtosecond lasers and the materials induces localized modification in the laser-affected region. When combined with wet or dry etching, modified areas can be selectively etched during micro-/nanoprocessing. This dual-process approach enhances precision and control, making femtosecond laser ablation an indispensable tool in the fabrication of intricate and high-precision three-dimensional micro-/nanostructures.

### 2.2.1. Femtosecond Laser Direct Writing Technology

For the processing of hard materials such as quartz, silicon, and metals, high-energy-density lasers are typically employed. This involves utilizing femtosecond laser ablation technology to scan and remove material, achieving the fabrication of intricate three-dimensional micro-/nanostructures. The laser beam is focused on the material's surface, and the processed material is placed on a three-dimensional platform. The computer program controls the motion of the three-dimensional platform. Therefore, the quality of the optical components depends on the precision, flatness, and stability of the platform's motion. Liu et al., utilized femtosecond laser direct writing technology to create positive spherical microlenses on a glass surface [88]. During the processing, a selective removal approach was first applied using a spiral scanning method to obtain the spherical cap structure. The removal of annular regions was then carried out layer by layer from top to bottom, as shown in Figure 8a. Debris deposited on the cap structure was cleaned using airflow, followed by a special smoothing treatment to reduce the number of processing layers and enhance processing quality, as illustrated in Figure 8b. Subsequently, a meridional arc smoothing process was employed to achieve microlenses with a desirable appearance. However, as a point-by-point scanning processing technique, processing efficiency becomes a significant concern when fabricating large-area, three-dimensional structures. Yong et al., addressed this challenge by using a high-speed femtosecond laser scanning method to rapidly produce a large-area concave MLA [89], as depicted in Figure 8c. Each microlens could be formed by a single femtosecond laser pulse. Within 50 min, approximately 2.78 million microlenses, with diameters of 8.68 µm and depths of 0.95 µm, were fabricated on a $2 \times 2$ cm$^2$ polydimethylsiloxane (PDMS) film. The mechanical pressure generated by the expansion of the laser-induced plasma on molten PDMS and the longer solidification time of PDMS contributed to the formation of high-quality microlenses. The fabricated microlenses demonstrated excellent optical performance, and the diameter and depth could be conveniently adjusted by varying the femtosecond laser power.

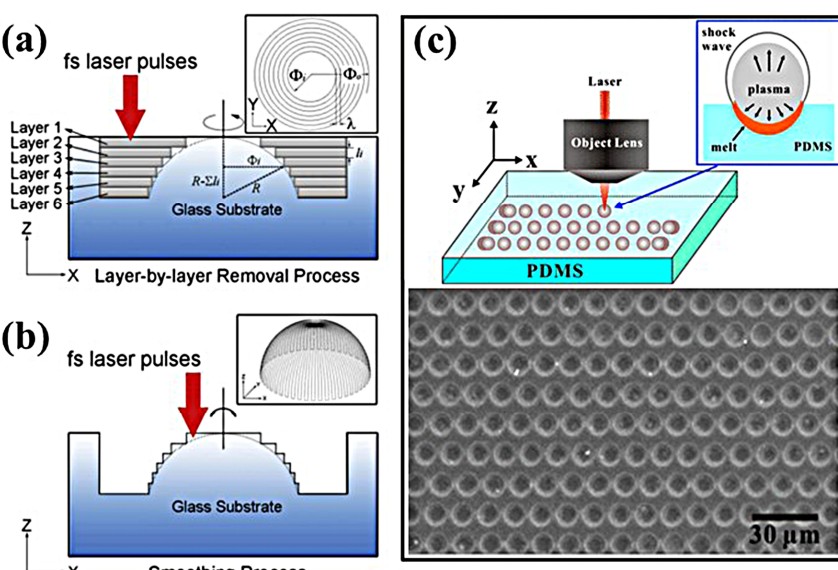

**Figure 8.** (**a**) The process of layer-by-layer removal. The shaded areas represent the regions to be removed by the laser pulses. The material is selectively eliminated through a spiral scanning path, progressing from layer 1 to layer 6. (**b**) Schematic of the fabrication process. Figure reproduced from (**a**,**b**) ref. [88], Elsevier. (**c**) The inset shows the formation mechanism of a microlens irradiated by a single-femtosecond laser pulse. The following image is a SEM image of the fabricated MLA. Figure reproduced from (**c**) ref. [89], American Chemical Society.

Table 2 provides the parameters for processing compound eyes in these two works. For femtosecond laser direct writing technology, the laser–material interaction region is located

at the material interface. Any material that absorbs a sufficient amount of energy and lacks interface constraints can undergo the ablation removal process. Therefore, the laser ablation processing mode can be applied to work with any material found in nature. However, material ablation removal is a dynamic process with unstable factors, such as dynamic changes in the ablation interface and the generation of ablation debris. On the one hand, these factors hinder the transmission of laser energy to the working area. On the other hand, they disrupt the state where the processing material lacks interface constraints. All these factors contribute to a cumulative impact on subsequent direct writing and ablation. Consequently, current femtosecond laser direct writing technology does not possess stable three-dimensional processing capabilities.

**Table 2.** Parameters of compound eyes fabricated by femtosecond laser direct writing technology.

| Year | No. of Ommatidium | Ommatidium Diameter (μm) | Ommatidium Height (μm) | FOV (°) | Laser Parameters | Materials | References |
|------|-------------------|--------------------------|------------------------|---------|------------------|-----------|------------|
| 2009 | 1 | 48 | 13.2 | - | 800 nm, 30 fs, 1 kHz | optical glass | [89] |
| 2013 | 2.78 million | 8.68 | 0.95 | - | 800 nm, 50 fs, 1 kHz | PDMS | [90] |

### 2.2.2. Femtosecond Laser Wet Etching Technology

Femtosecond laser wet etching stands out as a cutting-edge laser microfabrication process that has gained considerable traction in recent years. Essentially, this process hinges on the interaction between femtosecond lasers and materials. When femtosecond lasers induce material modification within a localized region, the chemical reactivity of the modified material surpasses that of the unmodified area. Consequently, during subsequent chemical wet etching, the modified material undergoes rapid removal [84,90–96]. The amalgamation of chemical wet etching with femtosecond laser processing has significantly enhanced efficiency, resulting in smoother surfaces. This technique has become a focal point of research interest [87,97–99], particularly in the realm of compound eye fabrication. Its innovative approach holds promise for advancing precision and quality in microfabrication processes, reflecting a noteworthy evolution in laser-based manufacturing technologies. Chen et al., utilized this method to manufacture large-area concave MLAs on silica glass [100]. Through laser irradiation and hydrofluoric acid etching processes, they prepared extensive rectangular and hexagonal arrays of concave microlenses with diameters less than 100 micrometers in a few hours, as shown in Figure 9a,b. The fabricated microlenses exhibited excellent surface quality and uniformity. In comparison to traditional processing techniques, this method is a maskless technology that allows for flexible control over the dimensions and shapes of the microlenses by adjusting parameters such as pulse energy, pulse count, and etching time. As crucial optical devices, artificial compound eyes are susceptible to contamination from airborne water droplets or similar pollutants [101,102]. To address this issue, Li et al., employed a combination of femtosecond laser wet etching and femtosecond laser direct writing to create superhydrophobic MLAs [103], shown in Figure 9c. The structure comprises MLAs and surrounding rough areas [Figure 9d]. The rough areas around each microlens were generated through subsequent femtosecond laser direct writing processes, resulting in the prepared MLAs exhibiting superhydrophobicity [104–107]. Water droplets can easily roll off the surface of the fabricated MLAs. By ingeniously combining smooth microlenses with the surrounding rough microstructures, the fabricated MLAs demonstrated excellent optical imaging capabilities and self-cleaning abilities. This flat compound eye structure holds potential applications in endoscopes, solar cells, and various optical systems frequently used in outdoor environments.

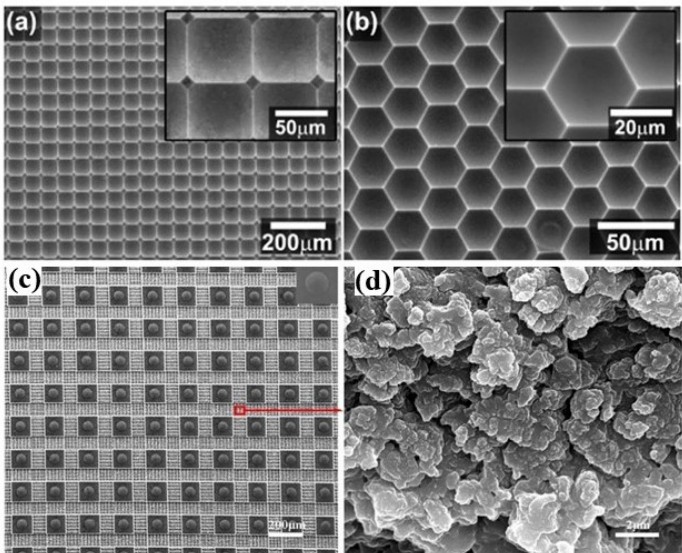

**Figure 9.** SEM images of (**a**) the rectangular MLA and (**b**) hexagonal MLA. (**c**) SEM image of the fabricated PDMS MLA (inset shows a single microlens). Figure reproduced from (**a**,**b**) ref. [100], Optical Society of America. (**d**) Magnified image of the laser-induced microstructures. Figure reproduced from (**c**,**d**) ref. [103], Wiley-VCH.

As exploration into artificial biomimetic compound eye structures progresses, recent breakthroughs have unveiled a curved biomimetic artificial compound eye that closely emulates its counterparts in the natural world. Deng et al., pioneered an innovative process, shown in Figure 10a, by seamlessly integrating femtosecond laser wet etching and thermal deformation to craft these intricate structures [108]. The fabrication journey begins with femtosecond laser wet etching, meticulously creating a concave compound eye lens array. Subsequently, a planar convex compound eye is created through the thermal imprinting of polymethyl methacrylate (PMMA) onto the existing structure. The final touch involves thermal pressing of this array onto a hemispherical mold, softening it to attain the desired curved structure. Upon cooling, a robust curved compound eye lens is formed. These artificial compound eyes boast hexagonal, rectangular, or irregular microlenses, each with nuanced shapes and sizes [Figure 10b,c]. The microlenses average approximately 24.5 μm in diameter for each small eye, amounting to a grand total of 30,000 small eyes [Figure 10d]. Remarkably, the field of view for a single artificial compound eye can extend up to 140°, and when paired with another eye, the collective field of view surpasses 180°, mirroring the panoramic perspective observed in natural compound eyes.

Beyond artificial compound eye structures optimized for high transmittance in the visible light spectrum, there have been notable advancements in the development of MLAs designed for the infrared (IR) wavelength range. Liu et al., pioneered the use of a femtosecond laser-induced wet etching technique to meticulously craft high-precision plano-concave microlens templates on a BK7 glass substrate [109]. Employing nanoimprinting technology, they successfully transferred the intricate microstructure array morphology onto an infrared polymer sheet, yielding a cutting-edge planar infrared MLA with superior performance, which is shown in Figure 11a. The specific processing technique involved the precise focusing of femtosecond laser pulses on a BK7 glass substrate, utilizing a deposition dose of 135 kJ/cm$^2$ and an irradiation interval of 150 μm. Following the laser-induced treatment, a sequence of chemical etching and polishing procedures was implemented, employing a 5% HF solution at room temperature to meticulously shape a rigid template. The choice of infrared PMMA material, characterized by a refractive index of 1.49 in the near-infrared range and a Vicat softening temperature of 107 °C, played a crucial role. Subsequently, an MLA device was replicated under a pressure of 87 kPa, resulting in a device with a well-defined surface morphology, exemplified in Figure 11b,c. The imaging

performance of the device underwent quantitative analysis using the Modulation transfer function (MTF) [Figure 11d], revealing a resolution in the infrared range that exceeded 100 lp/mm. Furthermore, both passive and active infrared imaging measurements were conducted using a halogen lamp as a target, showcasing the device's clear resolution of intricate details, such as a filament with a diameter of 0.1 mm. This innovative method introduces a novel approach to fabricating intricate three-dimensional micro-/nanostructures tailored to the infrared spectrum.

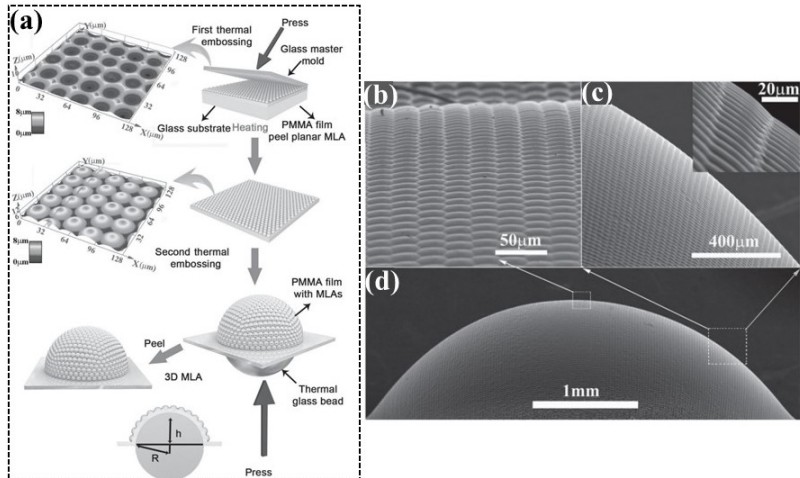

**Figure 10.** (**a**) The process schematic for manufacturing artificial compound eyes using femtosecond laser wet etching and thermal embossing. SEM images of the top (**b**) and side (**c**) of the artificial compound eye. (**d**) A high-power microscope view of the artificial compound eye. Figure reproduced from (**a**–**d**) ref. [108], Wiley.

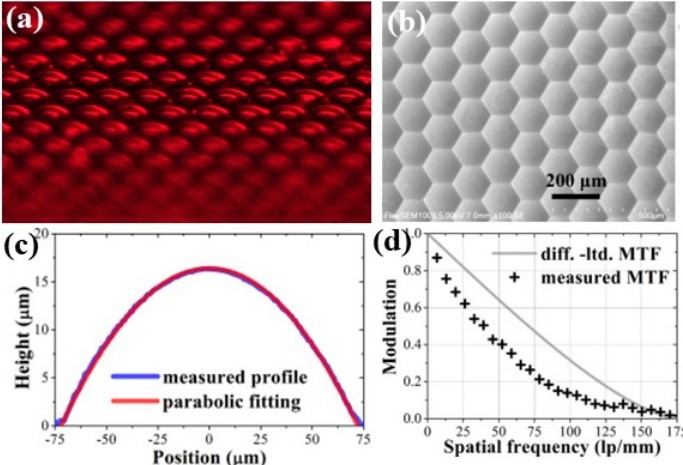

**Figure 11.** (**a**) The 3D morphology of the infrared micro-lens array surface. (**b**) Scanning electron microscope image of the device's surface morphology. (**c**) Cross-sectional morphology of a micro-lens unit. (**d**) MTF of the imaging unit. Figure reproduced from (**a**–**d**) ref. [109], Optical Society of America.

Accuracy when transforming a planar micro-lens array into a curved structure is paramount for the success of the two methods mentioned above. However, during the conversion process from a planar to a curved surface, unavoidable deformations arise due to differing curvatures at various positions, thereby affecting the uniformity of the sub-eyes within the artificial compound eye structure. An alternative fabrication approach entails directly manufacturing a micro-lens array on a spherical substrate. Wang et al., employed a concave glass lens, illustrated in the Figure 12a, as the substrate and harnessed a femtosecond laser-induced wet etching method to craft a three-dimensional concave lens

structure [110]. Following this, a two-step hot embossing technique was implemented to mold it with chalcogenide glass in order to obtain a biomimetic three-dimensional compound eye, as shown in Figure 12b. The intricate processing sequence began with the precise focus of femtosecond laser pulses, featuring a wavelength of 800 nm and a pulse width of 50 fs, onto the surface of a K9 concave glass. Employing a laser power of 17 mW and an irradiation time of 500 ms per pit, thousands of hexagonal pits, spaced at 90 μm intervals, were selectively ablated. Subsequent to the laser-induced treatment, a series of chemical etching and polishing steps ensued, involving the use of an 8% hydrofluoric acid solution to sculpt the K9 glass compound eye template. The chalcogenide glass $(Ge_{20}Sb_{15}Se_{65})$ material, possessing a specific softening temperature, was strategically placed at the center of the untreated K9 concave glass to create a preform under a pressure of 142 kPa. The preform was then vertically positioned at the core of the K9 glass artificial compound eye template, and the precise shaping process was replicated to fabricate the infrared artificial compound eye structure, mirroring the steps of the initial procedure. This compound eye incorporated 6000 small eyes arranged in concert, each boasting a diameter of 88 μm and a concave height of 11 μm [Figure 12c]. Moreover, the compound eye structure exhibited exceptional active imaging performance, achieving a remarkable resolution of up to 20.16 lp/mm [Figure 12d]. Subsequent to infrared passive imaging tests, the infrared compound eye showcased outstanding performance in the realm of infrared thermal imaging [Figure 11e], displaying a high transmittance of 60–70% in the range of 2.5 to 15 μm. The infrared compound eye accomplished large field-of-view imaging at 60°, signaling promising prospects for applications such as large-field imaging, infrared thermal imaging, and three-dimensional motion detection. The challenge of this approach lies in the continuous need to adjust the laser focusing position during processing. Specifically, when machining concave lenses, the focal point shifts downward with a decreasing radius, necessitating a substantial number of experiments and precise adjustments to achieve the desired outcome.

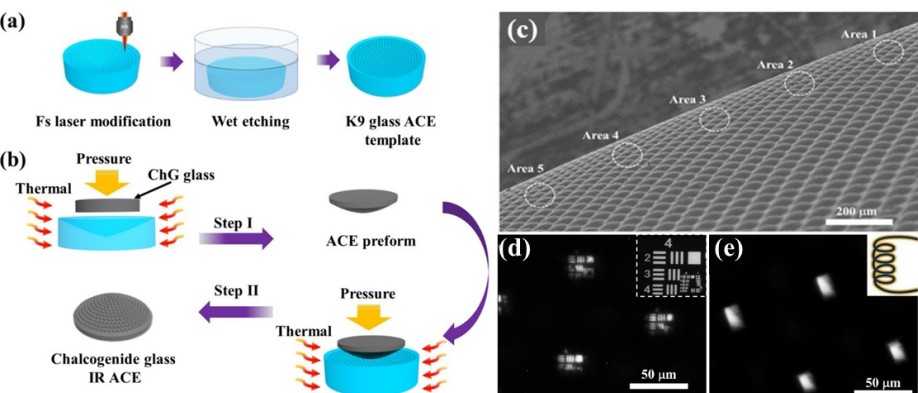

**Figure 12.** (**a**) Schematic diagram of the preparation of the K9 glass artificial compound eye template using femtosecond laser wet etching. (**b**) Schematic diagram of the "two-step" precision glass molding method for preparing the artificial compound eyes. (**c**) SEM image of artificial compound eyes. (**d**) Imaging of the USAF 1951 resolution test chart using an infrared compound eye. The inset shows the USAF 1951 resolution test chart. (**e**) Infrared passive imaging amplification of the heated tungsten wire after being powered on. The inset shows the infrared image of the tungsten wire. Figure reproduced from (**a**–**d**) ref. [110], Wiley.

Table 3 presents the parameters for processing compound eyes in these works. Wet-etching-assisted femtosecond laser modification technology has been widely employed in the fabrication of certain artificial compound eye structures. However, this method faces challenges in terms of processing materials with corrosion resistance. Wet etching of crystalline materials is typically based on crystal orientation, leading to distortions in the designed structure. Achieving isotropic chemical wet etching requires a mixture solution with complex and precise component ratios, adding to the process's requirements.

**Table 3.** Parameters of compound eyes fabricated using femtosecond laser wet etching technology.

| Year | No. of Ommatidium | Ommatidium Diameter (μm) | Ommatidium Height (μm) | FOV (°) | Laser Parameters | Materials | References |
|------|------|------|------|------|------|------|------|
| 2010 | ~20,000 | 67.05 rectangular 30.54 hexagonal | 10.68 rectangular 3.35 hexagonal | - | 800 nm, 30 fs, 1 kHz | silica glass | [101] |
| 2019 | ~2500 | 49.96 | 7.46 | - | 800 nm, 50 fs, 1 kHz | K9 glass concave MLAs/PDMS convex MLAs | [104] |
| 2016 | ~30,000 | 24.8 | 4.5 | 140 | 800 nm, 50 fs, 1 kHz | K9 glass concave MLAs/PMMA convex MLAs | [109] |
| 2019 | ~4400 | 150 | 16 | - | 800 nm, 50 fs, 1 kHz | K9 glass concave MLAs/NIR PMMA convex MLAs | [110] |
| 2022 | ~6000 | 88 | 11 | 60 | 800 nm, 50 fs, 1 kHz | K9 glass concave MLAs/Chalcogenide glass convex MLAs | [111] |

### 2.2.3. Femtosecond Laser Dry Etching Technology

Dry etching technology stands as the most mature and extensively utilized etching process in the industry. In essence, any etching technique that circumvents chemical solution corrosion is labeled as dry etching [111–114]. Owing to its commendable controllability, high precision, and mass etching capabilities in a production line, dry etching has evolved into a pivotal technology in semiconductor processes. Liu et al., pioneered the concept of dry-etching-assisted femtosecond laser processing technology, a two-step approach encompassing femtosecond laser ablation to generate modified regions and plasma etching (ICP) [115], as shown in Figure 13a,b. Initially, a femtosecond laser beam, boasting a wavelength of 800 nm and a repetition frequency of 1 kHz, was precisely focused on the surface of a silicon wafer, creating a microhole array on the silicon's surface. Subsequently, the irradiated silicon wafer underwent etching using inductively coupled plasma in sulfur hexafluoride (SF6) gas. The diameter of the concave structure expanded with the prolonged etching time, culminating in the formation of concave lenses, depicted in Figure 13c,d. Furthermore, by fine-tuning the laser power, pulse count, and etching duration, the diameter and height of the structures post-dry etching could be systematically adjusted.

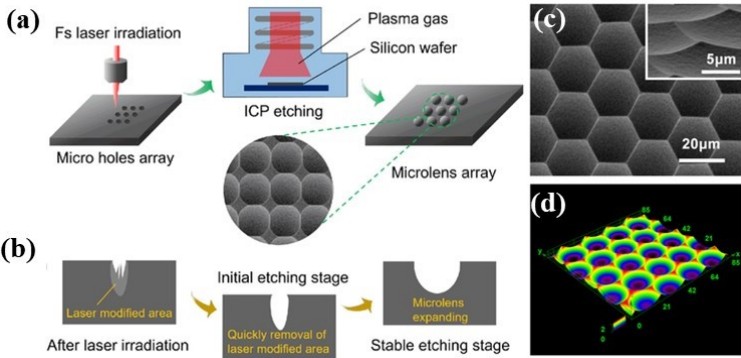

**Figure 13.** (**a**) Schematic diagram of the manufacturing process for a silicon MLA. (**b**) Schematic diagram illustrating the evolution of microlens formation. (**c**) SEM of the closely packed silicon concave MLAs. (**d**) The 3D profiles of microlenses. Figure reproduced from (**a–d**) ref. [115], Wiley.

In addition to silicon, high-power plasma etching has emerged as a versatile tool for processing other robust materials, including superhard materials, for three-dimensional manufacturing. Sapphire ($Al_2O_3$ single crystal), boasting a Mohs hardness of up to 9, ranks as the third hardest material in nature, showcasing exceptional wear resistance [116]. Additionally, sapphire exhibits high-temperature resistance, with a melting point soaring to 2040 °C. Leveraging these attributes, optical devices based on sapphire demonstrate promising applications in cutting-edge fields [117–120]. Sapphire's unique physical and chemical properties position it as a potential window or substrate material for LEDs and metasurfaces, given its high transparency across a broad spectral range from UV to mid-IR (0.18–4.5 μm). Liu et al., introduced a dry-etching-assisted femtosecond laser processing method for the rapid fabrication of artificial compound eyes on curved sapphire substrates, resulting in an efficiency improvement of over two orders of magnitude and a reduced processing time of 7 h, including 3 h for etching [121]. The processing steps are shown in Figure 14a. The morphology of the compound eyes can be finely tuned by adjusting laser energy and the number of pulses, offering a focal length range of 10 to 100 μm. Sapphire concave compound eyes, due to their high hardness and thermal stability, serve as ideal high-temperature hard casting templates. This enables the replication of convex compound eyes onto mineral glass materials such as K9 glass [Figure 13b]. Employing high-temperature casting replication technology, a fully glass compound eye comprising 190,000 ommatidia was successfully manufactured [Figure 14c,d]. Glass compound eyes boast attributes such as a wide field of view, exceeding 90°, and a high filling factor (≈100%). The diameters and heights of the sapphire concave microlenses can be precisely manipulated by adjusting the laser energy, pulse count, and etching time. The outcomes underscore that high-temperature casting replication of superhard materials offers a groundbreaking avenue for crafting three-dimensional micro-/nanostructures on hard materials.

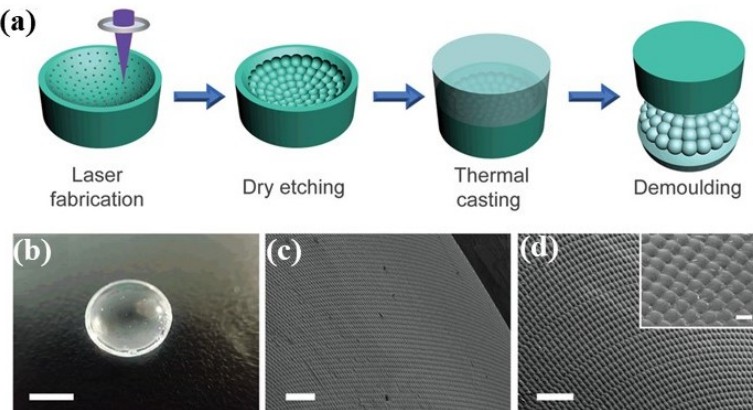

**Figure 14.** (**a**) Schematic diagram of the production process for a sapphire concave compound eye template and a K9 glass compound eye. (**b**) Photographic image of the K9 compound eye. (**c**) SEM image of the K9 glass compound eye. (**d**) Magnified SEM image with an insert showing a locally enlarged SEM image. Scale bars are 5 mm (**b**), 100 μm (**c**), 100 μm (**d**), and 20 μm (insert in (**d**)). Figure reproduced from (**a**–**d**) ref. [121], Wiley.

Table 4 provides the parameters for processing compound eyes in these works. Dry etching provides precise control over structural dimensions by adjusting plasma parameters to achieve specific etching profiles. However, it can sometimes result in surface damage due to high-energy ion bombardment, leading to defects like lattice damage or the formation of amorphous layers, thus adversely affecting device performance. Wet-etching-assisted femtosecond laser modification offers advantages in terms of manufacturing three-dimensional microchannels within hard materials, while dry etching-assisted femtosecond laser modification excels in surface processing. In certain scenarios, a combination of both techniques can be employed to strike the optimal balance between contour control, surface quality, and other process parameters, such as etching rate and selectivity.

**Table 4.** Parameters of compound eyes fabricated using femtosecond laser wet etching technology.

| Year | No. of Ommatidium | Ommatidium Diameter (µm) | Ommatidium Height (µm) | FOV (°) | Laser Parameters | Materials | References |
|------|------|------|------|------|------|------|------|
| 2017 | - | 20 | 1.6 | - | 800 nm, 100 fs, 1 kHz | silicon wafer | [116] |
| 2019 | ~190,000 | 20 | 11 | 90 | 800 nm, 100 fs, 1 kHz | sapphire concave MLAs/K9 glass convex MLAs | [122] |

## 3. Applications

In recent years, researchers have developed planar and curved artificial biomimetic compound eye structures. These biomimetic compound eye structures, as novel optical microstructures, have found extensive applications in various fields such as illumination engineering, radar systems, micro-aerial vehicles, compound eye cameras, night vision devices, and other defense and civilian equipment. In this section, we provide a brief overview of the application areas, including imaging, dynamic target capture, beam homogenization, and microfabrication.

The application of artificial compound eye structures and curved carrier optical microstructures in novel micro-optical devices has unparalleled advantages. Curved MLAs, with features such as wide-angle imaging, ultra-high resolution, and dynamic imaging, exhibit extensive application prospects. Inspired by the compound eye structures of insects such as moths, lobsters, and flies in the natural world, Song et al., combined flexible composite optical elements with deformable, thin silicon photodetectors to create an artificial compound eye camera [5], as shown in Figure 15a,b. This artificial compound eye camera offers a superior field of view compared to traditional cameras; reduces the camera's size; enhances spatial distribution; and, with excellent optical components, achieves imaging capabilities surpassing those of biological visual systems. The field of view of this biomimetic compound eye camera can be adjusted within the range of 140°–180° by changing the curvature radius of the elastic substrate through hydraulic means. Image reconstruction algorithms can restore captured images to their original three-dimensional objects. Floreano et al., integrated a three-layer structure comprising a micro-lens array, a silicon-based photodetector array, and a flexible printed circuit board to create a biomimetic compound eye system [122], as depicted in Figure 15c. The photodetector array was precisely cut into 42 columns, each containing 15 photodetector array units, then aligned and stacked with the micro-lens array (with alignment errors in the micrometer range). After curving it into a hemispherical shape, it was integrated with the flexible circuit board. The entire system weighed less than 1.7 g and had a volume of only 2.2 cm$^3$, a total power consumption of less than 0.9 W for 630 sub-eyes, and a field of view of 180° × 60°, as shown in Figure 15d.

Biomimetic compound eyes exhibit significant advantages in motion target detection. Hu et al., addressed the mismatch between the curved focal planes of biomimetic compound eye units and commercial planar image sensors by controlling the shape of the compound eye lens units to increase the depth of field for individual units [123], as shown in Figure 16a. The resulting micro-biomimetic compound eye lens had dimensions of 400 µm, containing 160 lens units capable of achieving wide-angle imaging within a 90° range. Experimental tests on the spatial position recognition and motion trajectory detection capabilities of this micro biomimetic compound eye structure revealed real-time reconstruction of the movement trajectory of a paramecium, as depicted in Figure 11a. Zheng et al., introduced a three-dimensional trajectory detection technique based on a curved compound eye [124]. The overall structure of this compound eye vision system is illustrated in Figure 16b, consisting of a curved compound eye lens, aperture, and top light cone. By integrating the compound eye system with a CCD camera, it rapidly determined the three-dimensional positions of objects by analyzing the position and intensity distribution of light, achieving efficient and accurate localization.

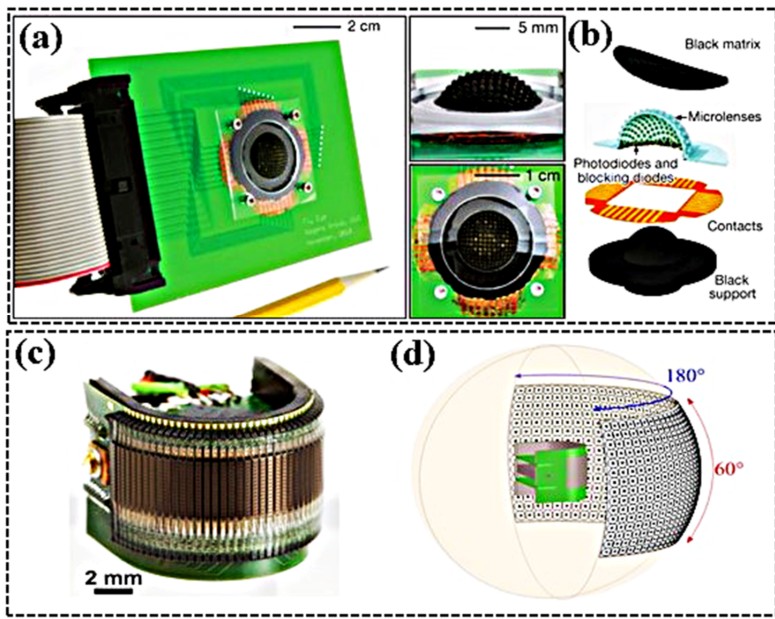

**Figure 15.** (**a**) A photograph of the complete camera mounted on a printed circuit board, with insets featuring a tilted view (upper inset) and a top-down view (lower inset). (**b**) Exploded view of the system components: a black silicon perforated sheet (black matrix), a hemispherical array of micro-lenses and photodiodes/avalanche diodes, thin-film contact points for external interconnection, and a hemispherical black silicon support substrate. Figure reproduced from (**a**,**b**) ref. [5], Nature. (**c**) Image of the curved camera. The entire device has a volume of 2.2 cm$^3$, a weight of 1.75 g, and a maximum power consumption of 0.9 W. (**d**) Illustration of the panoramic FOV of the fabricated prototype. Figure reproduced from (**c**,**d**) ref. [122], National Academy of Sciences.

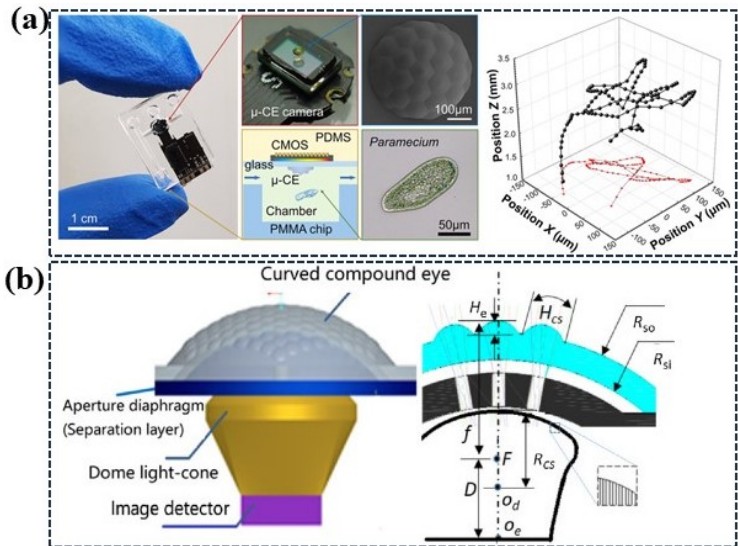

**Figure 16.** (**a**) Photograph of the on-chip camera system (left), the μCE camera (top middle), and the SEM image of the compound eye (top right). The insets include a schematic illustration of the working mechanism of a paramecium (bottom to middle) and a microscopic image of a paramecium. (**b**) The reconstructed 3D trajectory of the Paramecia. (Figure reproduced from (**a**,**b**) ref. [123], Nature. (**b**) Curved bionic compound-eye lens. Figure reproduced from (**b**) ref. [124], Optical Society of America.

The refractive concave MLA has immense application value in micro-optical systems and structured light fields, particularly in applications such as beam homogenization, where the divergent characteristics of concave lenses can prevent the generation of "hot

spots" in the laser's transmission path. Deng et al., utilized single-pulse femtosecond laser-assisted chemical wet etching to fabricate a double-sided MLA [125] consisting of irregularly arranged concave MLAs on both sides with specific rotation angles, as shown in Figure 17a. Simulation and experimental results indicated that the homogenization performance was optimal when the rotation angle of the MLAs on both sides was 60° [Figure 17b]. Additionally, the MLA can function as a parallel lens group for achieving high-throughput micro-/nanoprocessing. Liu et al., employed a combination of femtosecond laser wet etching and hot embossing techniques to fabricate an infrared PMMA MLA [109]. Using the damage probability method, the laser damage thresholds for "single-shot" and "multiple-shot" conditions of the infrared PMMA material under near-infrared ultrashort pulses (wavelength of 1030 nm, pulse width of 350 fs) were determined to be 2 J/cm$^2$ and 0.7 J/cm$^2$, respectively, as shown in Figure 17c. Focusing the collimated femtosecond laser beam through the infrared MLA, the focal spot size after the focusing device was 3.1 µm, with a fluence density of 2.1 J/cm$^2$, exceeding the laser damage threshold of the polymer. This allowed for the writing of patterns on the polymer target material, demonstrating uniform pattern details and spacing (Figure 17d), showcasing the practical application potential of the microlens.

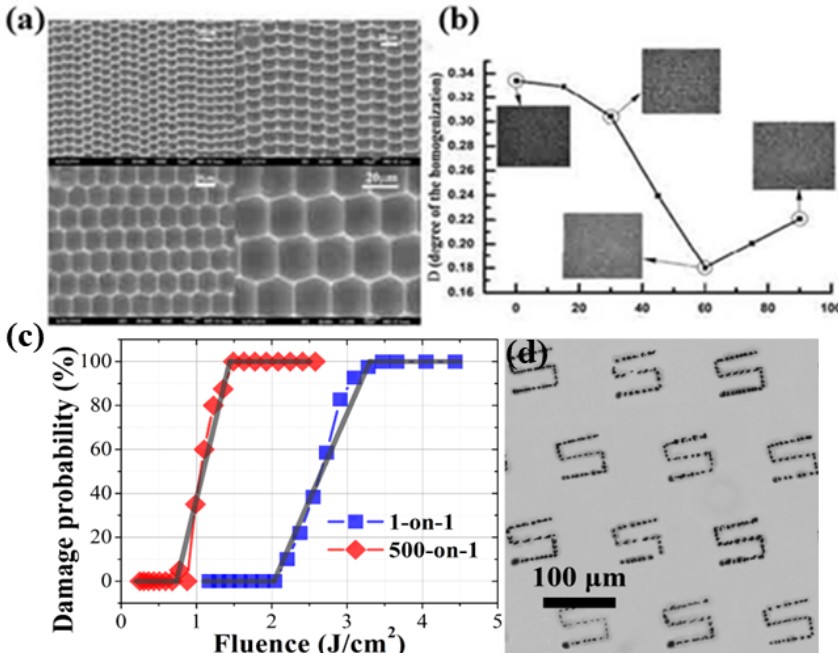

**Figure 17.** (**a**) The SEM images of the fabricated microlenses. (**b**) A change of the degree of homogenization (D) with the angles (θ) between scans. Inset: Illumination patterns of the diffuser at different scanning angles. Figure reproduced from (**a**,**b**) ref [125], IEEE Xplore. (**c**) Relationship between laser ablation probability and incident energy for the variable region data (solid gray line) to determine the damage threshold. (**d**) laser micro-etching pattern of the MLA device (left). Figure reproduced from (**c**,**d**) ref. [109], Optical Society of America.

## 4. Conclusions

In the early 20th century, scientists embarked on the development of biomimetic compound eye systems. Initially constrained by planar fabrication processes, the emphasis was on creating planar biomimetic compound eye devices that replicated the multi-aperture imaging pattern of compound eyes. With the robust evolution of computer algorithms, researchers delved into methods to enhance the resolution of planar biomimetic compound eyes and continuously explored their practical applications. Planar biomimetic compound eye systems have exhibited significant potential in areas such as depth detection and three-dimensional display. Given the primary advantage of biomimetic compound eyes,

which lies in the expansive field of view provided by the curved surface, curved biomimetic compound eyes have emerged alongside the flourishing of planar counterparts. Currently, fabrication processes for curved artificial compound eyes include femtosecond laser micro-fabrication technology, as well as techniques like self-assembly, thermally induced reflow deformation, ultra-precision mechanical machining, and others. Despite various processing techniques achieving the fabrication of curved microstructures, challenges persist in the manufacturing processes. The utilization of self-assembly of micro/nanospheres made of polymer molecules can form the required array structure at the nanometer level, representing a promising method for manufacturing the smallest artificial compound eyes. However, the shape of the spheres is susceptible to deformation due to excessive compression, and aggregation may not be sufficiently tight. Therefore, ensuring the sphericity of the spheres remains a challenge to be addressed. Currently, the most commonly used method is the thermal reflow method due to its simplicity and cost-effectiveness. It can be combined with various deformation methods such as negative pressure deformation, gas-assisted deformation, and microfluidic devices, among others. However, curved artificial compound eyes prepared by thermal reflow tend to have larger dimensions compared to those in nature. On the other hand, single-point diamond micro-machining may result in micrograting structures on the surface of the micro-lens array units due to larger tool spacing, leading to higher surface roughness.

Femtosecond laser processing technology is increasingly asserting its unique advantages in crafting artificial biomimetic compound eye structures, leveraging its precision, programmable design, and three-dimensional processing capabilities. This review provides a concise overview of the advancements in femtosecond laser-prepared compound eyes, emphasizing two pivotal femtosecond laser processing techniques. (1) The first is femtosecond laser additive manufacturing, specifically referring to femtosecond laser two-photon polymerization. This bottom-up preparation method utilizes the dual-photon absorption effect when the femtosecond laser interacts with polymers, making it suitable for soft polymers such as photoresist and proteins. Fine-tuning of laser parameters and polymer materials allows for the precise fabrication of highly detailed three-dimensional structures. (2) Femtosecond laser subtractive manufacturing, primarily referring to femtosecond laser ablation, utilizes the ultra-high energy of a femtosecond laser to selectively remove or modify the substrate. This approach is applicable to hard substrates, including semiconductors, metals, and dielectric materials. Further processing through wet or dry etching ensures surface smoothness, making it suitable for the fabrication of complex three-dimensional structures. Both femtosecond laser processing methods possess the capability to create miniaturized, intricate, three-dimensional biomimetic structures, establishing femtosecond laser technology as a key player in the next generation of artificial compound eye fabrication. Its advantages, including high processing precision, strong controllability, and applicability to various material types, provide robust support for the development of biomimetic compound eyes.

However, there are still some obstacles that need to be overcome. (1) Further improvement of processing efficiency: In actual manufacturing, there is a challenging balance between the dimensions, unit numbers, surface precision of devices, and processing efficiency. For example, two-photon polymerization technology, as a point-by-point scanning processing method, has a long processing time, relatively low efficiency, and is not conducive to practical industrial production [74,78,126]. While femtosecond laser ablation technology can achieve the preparation of complex three-dimensional micro-/nanostructures on hard materials, processing hard materials typically requires high laser energy density, which makes it challenging to increase processing rates. It is essential to explore innovative strategies aimed at enhancing the fabrication efficiency of femtosecond laser modification [109,127]. For example, parallel processing technologies such as laser interference and spatial light modulators are employed for multi-beam processing and three-dimensional focal field scanning to enhance processing efficiency [128–130]. (2) High-precision manufacturing challenges for complex three-dimensional biomimetic compound eyes include

the following: Biological compound eyes are composed of numerous facets, each with nearly perfect geometric shapes. The size, spacing, arrangement, and effective quantity of each facet directly impact the imaging quality, responsiveness, and field of view of the compound eye. As a precise optical component, a biomimetic compound eye needs to have a three-dimensional surface configuration comparable to that of an biological compound eye. The array of small eye lenses not only needs to be closely arranged, but also requires external contours to strictly adhere to specific function curves [131]. The preparation of such a high-precision three-dimensional compound eye structure poses a severe challenge to traditional manufacturing processes. Certainly, prior to fabrication, one can utilize relevant software (such as ZEMAX) to simulate compound eye parameters, calculate the required specifications, and produce an artificial compound eye structure comparable to that of an insect compound eye [123]. (3) Integration challenges of optoelectronic functional components: For macroscopic biomimetic compound eye systems, assembly is typically achieved by assembling various discrete components. Traditional preparation methods involve the independent processing and fabrication of each component, followed by precision assembly. This method not only increases the volume of the optoelectronic system, but also introduces information loss due to component coupling, thereby diminishing the performance of the compound eye. This approach is clearly unsuitable for miniature compound eyes. When the size of the compound eye is in the millimeter or even hundred-micrometer range, achieving precise alignment and assembly of multiple components becomes extremely challenging. To address the integration challenges, there are several approaches: (a) utilize light path folding to converge images distributed on curved lenses onto a flat surface. This includes techniques such as fiber optic bending and lens bending; or (b) develop flexible image sensors that can directly capture images formed by the curved compound eye.

The potential applications and developmental possibilities of curved artificial compound eyes are vast. These future artificial compound eyes have the capability to mimic natural vision more effectively and adapt to a wide range of scenarios. By utilizing innovative materials, they can operate in different wavelengths, including infrared and ultraviolet, broadening their usability across diverse environments. Concurrently, improvements in the optical and mechanical performance of biomimetic compound eyes enhance their overall stability and durability. The integration of intelligent technology and artificial intelligence algorithms empowers biomimetic compound eyes to process and interpret visual information intelligently, allowing them to comprehend and respond to complex scenes. An illustrative example is the integration of artificial compound eyes with CCD cameras, forming intelligent electronic panoramic cameras. With their expansive field of view, these eyes find applications in medicine, contributing to the development of endoscopes for enhanced monitoring of internal body structures. Moreover, the compound eye structure holds the potential to be incorporated into unmanned aerial vehicles, small robots, or automobiles for navigation and surveillance purposes [132]. Despite successful advancements in producing artificial compound eyes using various technologies, imaging systems still grapple with challenges such as large volume and complex manufacturing processes. Future research efforts should concentrate on seamlessly integrating curved micro-lens arrays into compact devices; achieving commercialization; and propelling advancements in panoramic imaging, 3D information extraction, biomedical applications, and navigation and positioning technologies.

**Author Contributions:** Conceptualization, F.C. and F.Z.; validation, H.X. and F.Z.; formal analysis, H.X. and F.Z.; investigation, F.Z. and Q.Y.; data curation, Y.L. and G.D.; writing—original draft preparation, F.Z.; writing—review and editing, F.Z. and G.D.; supervision, F.C. and Q.Y.; project administration, F.C. All authors have read and agreed to the published version of the manuscript.

**Funding:** This work is supported by the National Science Foundation of China under the Grant (nos. 12127806, 62175195), the International Joint Research Laboratory for Micro/Nano Manufacturing and Measurement Technologies, the Natural Science Foundation of Shandong Province (ZR2023QF107), the Doctoral Research Foundation of Liaocheng University (318052215).

**Institutional Review Board Statement:** The study did not require ethical approval.

**Informed Consent Statement:** Informed consent was obtained from all subjects involved in the study.

**Data Availability Statement:** Informed consent was obtained from all subjects involved in the study.

**Conflicts of Interest:** The authors declare no conflicts of interest.

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
