# Peer review of "Femtosecond Laser Microfabrication of Artificial Compound Eyes"

_photonics, doi:10.3390/photonics11030264_

Round 1

Reviewer 1 Report

Comments and Suggestions for Authors

Femtosecond laser processing technology shows great potential in the field of artificial bionic compound eye structures, because of its excellent precision, programmable design capabilities, and 3D processing properties. Particularly, the femtosecond laser micromachining can minimize damage to the surrounding area and facilitate ultra-high precision micro-nano fabrication. In this review paper, Chen et al. systematically introduced the operational principles of biological compound eyes, and also summarized the typical manufacturing methods for biomimetic compound eye structures. they also introduced the femtosecond laser technology for producing biomimetic compound eye. This review is very interesting and well-written, which will arouse a wide readership from photonics, materials, and physics. Therefore, I am pleased to recommend this manuscript for publication with minor revision. The following comments may help improve the manuscript.

1. In Figure 1, could the authors provide more examples of compound eyes in nature, not just the dragonfly?

2. When the abbreviation of MLA first appeared on page 2, authors should give its full name which is microlens array.

3. In Figure 8d, please remove the irrelevant Chinese character. 

Comments on the Quality of English Language

Minor editing of English language required

Author Response

Dear Reviewer,

Thank you very much for taking the time to review this manuscript. I would like to express my sincere gratitude for the insightful comments and feedback you provided on my submission. Your expertise and guidance have been immensely valuable in refining the quality of my work.

We have studied comments carefully and tried our best to improve the manuscript according to their suggestion. The responses to all comments are listed below point-by-point. The corresponding revisions in the paper are highlighted in the revised paper with a color change (the deleted text is marked with a strikethrough and in red, the added text is marked in blue). We hope the current version can meet the high demand of our journal.

Please find the enclosed revised manuscript, “Femtosecond Laser Microfabrication of Artificial Compound Eyes” by Fan Zhang et al. We confirm that this manuscript has not been published elsewhere and is not under consideration by another journal. Of course, all authors have approved the manuscript and agree with its submission to Photonics.

Response to reviewer's comments - Please see the attachment.

Thanks very much for your great helps.

Sincerely yours

Fan Zhang

Reviewer 2 Report

Comments and Suggestions for Authors

Fan Zhang et al. have summarized the application of femtosecond laser technology in the production of biomimetic compound eye. And this review concludes by highlighting the current challenges and presenting a forward-looking perspective on the future of this evolving field. It is well written and interest to the researchers in the related areas. I would consider the paper for publication after minor revisions are made according to the following specific comments:

1.      Why fs laser beam is used for fabrication of Artificial Compound Eyes? Could nanosecond laser or picosecond laser achieve the same results?

2.      The discussion on future prospects lacks depth. Provide a more nuanced analysis of potential advancements of Artificial Compound Eyes. Discuss emerging trends, ongoing challenges, and propose potential research avenues that could shape the field in the coming years.

3.      One significant drawback is the absence of the author's insights and perspectives. The manuscript reads more like a compilation of summaries from various papers rather than an expert synthesis of the literature. It is crucial to infuse the text with the author's critical viewpoints, potentially addressing gaps in current research, proposing new directions, highlighting methodological limitations in the existing studies, and providing a more engaging and enlightening narrative for the readers.

4.      For the study of femtosecond laser fabrication and its applications, the authors may refer these recent papers: Nanoscale, 2023,15, 11247; Nanoscale 15, 15708, 2023;

5.      For more perfection, several language mistakes could be revised.

Comments on the Quality of English Language

For more perfection, several language mistakes could be revised.

Author Response

(The authors gave the same response as above.)

Reviewer 3 Report

Comments and Suggestions for Authors

As the author is well aware, the main parameters of femtosecond lasers include pulse energy, pulse repetition rate, pulse duration, wavelength, focusing power (NA of the focusing optics), Focus spot shape (Gaussian or other), and spatial pulse spacing. It is important to tabulate each parameter in a review paper to encourage quick understanding by readers. Although the author classified various technologies into categories, it would be better to organize the data from the literature again targeting key parameters.

Please carefully check units, subscripts, spelling, etc.

Comments on the Quality of English Language

Check units, subscripts, symbols, spelling, spaces, etc. carefully.

Author Response

(The authors gave the same response as above.)

Round 2

Reviewer 3 Report

Comments and Suggestions for Authors

The authors have addressed all my previous concerns and I believe it can be published as is.